# Position: Zeroth-Order Optimization in Deep Learning Is Underexplored, Not Underpowered

**Sijia Liu** [* 1]  **Yicheng Lang** [* 1]  **Soumyadeep Pal** [1]  **Changsheng Wang** [1]  **Yancheng Huang** [1]  **Chongyu Fan** [1]
**James Diffenderfer** [2]  **Bhavya Kailkhura** [2]  **Yihua Zhang** [1]

## Abstract

Zeroth-order (ZO) optimization, learning from finite differences of function evaluations without backpropagation, has recently regained attention in deep learning due to its memory efficiency and applicability to gray- or black-box pipelines. Yet, ZO methods are often dismissed as fundamentally unscalable because of estimator variance and unfavorable query complexity. We argue that this conclusion might be misguided: **ZO optimization is underexplored, not underpowered**. We show that many perceived limitations stem from myopic development practices, most notably full-space, element-wise, estimator-centric designs. We articulate six positions spanning the algorithmic, systems, and evaluation stack. First, we revisit the feasibility boundaries of estimator-centric ZO methods through variance control, variance–query tradeoffs, and directional-derivative lenses. Then, we identify three underexplored opportunities: (i) subspace and spectral views of ZO that enable interpretable variance reduction with graceful query scaling, (ii) the forward-only nature of ZO as a systems advantage for communication-efficient, pipeline-friendly, and resource-constrained training, and (iii) the need to de-obfuscate ZO evaluations from task complexity. We strongly advocate rethinking ZO optimization around its unique strengths and acting accordingly, opening a viable path toward large-scale, system-aware, and resource-efficient learning with ZO optimization.

## 1. Introduction

Zeroth-order (**ZO**) optimization refers to the gradient-free counterpart of first-order (**FO**) optimization. That is, instead of accessing exact or stochastic FO gradients, ZO methods approximate descent directions using function-value–based gradient estimators (Spall, 1987; Duchi et al., 2015; Nesterov & Spokoiny, 2017; Flaxman et al., 2004; Liu et al., 2020). Although this function access may appear restrictive, ZO optimization is broadly relevant to deep learning for two reasons. First, ZO optimization offers a principled alternative to backpropagation (BP) for *efficient learning* (Malladi et al., 2023; Zhang et al., 2024c; Tan et al., 2025). By eliminating backward passes, ZO methods can substantially reduce activation memory, simplify execution pipelines, and enable training under hardware, system, or deployment constraints where BP is impractical or inefficient (Gu et al., 2021; Zhao et al., 2025c). Second, ZO optimization is indispensable for *black-box and gray-box learning* (Zhang et al., 2022; Chen et al., 2024; Ma et al., 2024; Ilyas et al., 2018; Sun et al., 2022), where FO gradients are unavailable, unreliable, or prohibitively expensive. This setting commonly arises in learning scenarios involving black-box components, such as simulators or ML-as-a-service (MLaaS) systems.

Prior to 2023, ZO optimization established its empirical credibility in deep learning through *input-level black-box applications*, most notably adversarial example generation (Chen et al., 2017; Ilyas et al., 2018) and input prompting (Sun et al., 2022; Tsai et al., 2020) under black-box models or query-only APIs. These successes positioned ZO optimization as a practical and competitive tool when gradient information is inaccessible. However, such applications typically operate in relatively low-dimensional input spaces (*e.g.*, images or token sequences), where the core scalability challenges of ZO optimization, such as gradient estimation variance and query cost, are largely suppressed.

A qualitative shift occurred with the emergence of *memory-efficient fine-tuning via ZO optimization* (Malladi et al., 2023), which moved ZO methods from input-space manipulation to the much higher-dimensional regime of *weight-level model training*. This shift has sparked a surge of ZO research since 2023, as shown in **Fig. 1(Left)**. By replacing

---

*Equal contribution [1]OPTML Lab, Michigan State University, USA [2]Lawrence Livermore National Laboratory, USA. Correspondence to: Sijia Liu <liusiji5@msu.edu>, Yicheng Lang <langyich@msu.edu>.

*Proceedings of the $43^{rd}$ International Conference on Machine Learning*, Seoul, South Korea. PMLR 306, 2026. Copyright 2026 by the author(s).

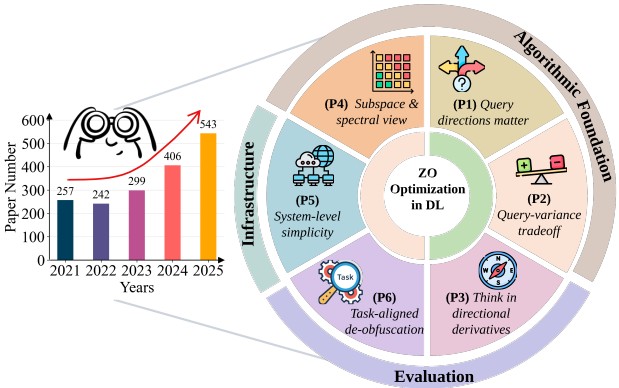

*Figure 1.* Schematic overview of this position paper. *(Left)* Publication trends (papers per year) for works with "ZO optimization" in the title in arXiv cs.AI and cs.LG (machine learning). *(Right)* Conceptual organization of our positioning points (P1–P6).

BP with forward-only passes, ZO optimization can fine-tune large pretrained models, while significantly *reducing memory overhead* (Zhang et al., 2024c). Thus, a growing literature now aims to scale ZO optimization for large-model fine-tuning, proposing increasingly sophisticated ZO gradient estimators and variance-reduction techniques (Liu et al., 2025b; Yu et al., 2025; Zhao et al., 2025a; Chen et al., 2025).

Beyond memory efficiency, another key advantage of ZO training is *flexibility*: it enables end-to-end optimization of hybrid systems composed of both ML and non-ML components, where differentiability is neither natural nor desirable. This capability is particularly relevant to emerging paradigms such as physics-informed neural networks (Raissi et al., 2019), digital twins (Mihai et al., 2022; Ma et al., 2024), and agentic AI (Acharya et al., 2025), whose pipelines often combine symbolic logic, simulators, tools, or environment interactions with neural modules. However, progress along this direction remains limited (Zhao et al., 2023b; Chen et al., 2024), largely because these use cases often require *training from scratch* with ZO optimization. As a result, existing efforts are mostly confined to small architectures (*e.g.*, CNNs) rather than large-scale models (Chen et al., 2024; Yue et al., 2025). This has also fueled skepticism in the community and raises a natural question: *Is ZO optimization fundamentally limited in deep learning, or have we simply not explored it in the right way?*

In this work, **we take the position that ZO optimization is underexplored, not underpowered.** We argue that many perceived limitations of ZO optimization stem *not* from intrinsic weaknesses of gradient-free learning, but from myopic development practices, most notably, an overemphasis on gradient-estimator–centric design in the element-wise parameter space. Existing approaches overlook broader algorithmic, system-level, and evaluation opportunities that are critical to advancing ZO optimization in deep learning.

In this work, we articulate **six positioning arguments (P1–P6)** spanning the algorithm–system–evaluation stack; see a schematic overview in **Fig. 1(Right)**. Specifically, **positions P1–P3** characterize the feasibility boundaries of current ZO optimization (see **Sec. 2**). From an *algorithmic* perspective, P1 and P2 are proposed to pinpoint the most critical design factor in existing studies–variance control in ZO gradient estimation. From an *evaluation* perspective, P3 highlights an often-overlooked feasibility diagnostic: the directional-derivative–based formulation can serve as a principled reference for assessing the difficulty of ZO optimization.

**Positions P4–P6** call for rethinking and advancing ZO optimization from complementary perspectives (see **Sec. 3**). From an *algorithmic* perspective, P4 challenges the prevailing practice of operating ZO methods in the original parameter space, which overlooks the substantial scalability advantages of *subspace* learning. From an *infrastructure* perspective, P5 advocates exploiting the simplicity of ZO forward-only execution as a systems advantage, including natural parallelism and communication efficiency, properties that are highly aligned with distributed and resource-constrained learning environments. From an *evaluation* perspective, P6 calls for disentangling the performance of ZO optimization from its associated *task complexity* to avoid obscured assessments of ZO's true optimization power.

Furthermore, we distill P1–P6 into a concise set of actionable calls in **Sec. 4** and contrast our position with several alternative viewpoints in **Sec. 5**.

**Conflict of Interest Disclosure**   The authors declare that they have no conflicts of interest.

## 2. Foundations and Feasibility Boundaries of ZO Gradient Estimator-Centric Design

Most existing ZO methods rely on query-based gradient estimators that approximate FO gradients using only function evaluations, enabling FO optimization frameworks to be reused in the ZO setting. In this section, we briefly review the foundations of this estimator-centric paradigm and argue that, even within it, several critical considerations have been overlooked and deserve closer attention.

Consider the minimization problem $\min_{\mathbf{x} \in \mathbb{R}^d} f(\mathbf{x})$, where $f(\mathbf{x})$ is an objective function defined over a $d$-dimensional optimization variable $\mathbf{x} \in \mathbb{R}^d$. A vanilla ZO gradient estimator for approximating the FO gradient $\nabla_{\mathbf{x}} f(\mathbf{x})$, known as a *randomized gradient estimator (RGE)*, is given by

$$\hat{\nabla}_{\mathbf{x}} f(\mathbf{x}) = \frac{f(\mathbf{x} + \mu \mathbf{u}) - f(\mathbf{x})}{\mu} \mathbf{u}, \quad \text{(RGE)}$$

where $\hat{\nabla}_{\mathbf{x}} f(\mathbf{x})$ denotes an estimate of the FO gradient, $\mathbf{u} \in \mathbb{R}^d$ is a random perturbation direction, *e.g.*, drawn from a standard Gaussian distribution (*i.e.*, $\mathcal{N}(\mathbf{0}, \mathbf{I}_d)$), and

$\mu > 0$ is the perturbation size. The formulation in (RGE) adopts a forward finite-difference scheme based on function evaluations, commonly referred to as the *one-sided* RGE. Alternatively, a *two-sided* RGE can be constructed using a central finite-difference scheme, $\frac{f(\mathbf{x}+\mu\mathbf{u})-f(\mathbf{x}-\mu\mathbf{u})}{2\mu}\mathbf{u}$. Replacing the FO gradient with RGE within an FO optimization framework yields the *basic* ZO optimization scheme, from which many existing variants are derived. For example, a ZO gradient descent (GD) or stochastic gradient descent (SGD) takes the form $\mathbf{x}_{t+1} = \mathbf{x}_t - \eta \hat{\nabla}_{\mathbf{x}} f(\mathbf{x}_t)$ at iteration $t$, where $\eta > 0$ is a proper learning rate.

Despite its deceptively simple form in (RGE), this estimator conceals several critical design choices that fundamentally shape the behavior of ZO optimization. We next articulate our first three positioning points (**P1**–**P3**), which are known but often overlooked, and which delineate the *feasibility boundaries* of ZO optimization.

> **(P1) Random direction choice matters.** The choice of the perturbation direction $\mathbf{u}$ plays a central role in controlling gradient estimation variance in ZO optimization.

**Argument for (P1).** A common choice for the random direction $\mathbf{u}$ is a standard Gaussian vector, $\mathbf{u} \sim \mathcal{N}(\mathbf{0}, \mathbf{I})$. The rationale is that RGE becomes an unbiased estimator of the gradient of the *Gaussian smoothing*-based objective $f_\mu(\mathbf{x}) \triangleq \mathbb{E}_{\mathbf{u}}[f(\mathbf{x} + \mu\mathbf{u})]$ (Nesterov & Spokoiny, 2017). In addition, when $\mathbf{u}$ is chosen as a symmetric $\pm 1$-valued Bernoulli random vector, the resulting estimator corresponds to the *simultaneous perturbation stochastic approximation* (SPSA) (Spall, 2002; LA & Bhatnagar, 2025).

Despite different choices of the perturbation vector $\mathbf{u}$, the resulting ZO gradient estimators remain coarse approximations whose variance scales proportionally with the parameter dimension $d$ (Liu et al., 2020). This intrinsic variance explosion constitutes a fundamental bottleneck to achieving high-precision and scalable ZO optimization.

To reduce the variance of ZO gradient estimates, it is advantageous to move beyond *isotropic Gaussian* perturbations and instead employ *structured designs* for the random direction $\mathbf{u}$. One representative approach is the *sparse RGE* (Liu et al., 2025b; Chen et al., 2024), which applies a pruning mask to a standard Gaussian perturbation vector, yielding a sparse direction $\hat{\mathbf{u}} = \mathbf{m} \odot \mathbf{u}$, where $\mathbf{m}$ is a binary mask indicating active parameters. This design reduces the effective dimensionality of the perturbation, and the estimator variance scales proportionally with the number of selected parameters. Another line of work introduces *preconditioning* into ZO gradient estimation. For example, *preconditioned simultaneous perturbation stochastic approximation* (PSPSA) (Zhao et al., 2025a;b; Seung et al., 2026) applies an approximate inverse Hessian to reshape the perturbation directions, thereby aligning random queries with the local geometry of the objective and improving estimation efficiency. However, both sparse and preconditioned perturbations introduce additional complexity in computation and query cost, which may violate (P2) discussed later. The former requires deciding which parameters to perturb, while the latter relies on estimating the Hessian, which is difficult.

> **(P2) Query-variance tradeoff.** Variance reduction in ZO optimization must be considered jointly with its induced query complexity; ignoring this tradeoff leads to misleading conclusions about scalability and feasibility.

**Argument for (P2).** The ZO gradient estimator is also implemented in a mini-batch form, where batching can arise from two sources: the collection of random perturbation directions $\{\mathbf{u}_i\}_{i=1}^n$ and the data samples $\{\omega_j\}_{j=1}^m$ used to evaluate the objective function. Here $\mathbf{u}_i$ and $\omega_j$ denote the $i$th random vector and $j$th sample, respectively. This extends RGE to its mini-batch variant,

$$\hat{\nabla}_{\mathbf{x}} f(\mathbf{x}) = \frac{1}{mn} \sum_{j=1}^{m} \sum_{i=1}^{n} \frac{f(\mathbf{x} + \mu\mathbf{u}_i; \omega_j) - f(\mathbf{x}; \omega_j)}{\mu} \mathbf{u}_i, \quad (1)$$

where $f(\mathbf{x}; \omega_j)$ denotes the function value at the data sample $\omega_j$. Although the mini-batch RGE incurs a higher function-query cost on the order of $O(mn)$ relative to the vanilla estimator in (RGE), this increased cost yields a reduction in ZO gradient variance, which scales as $O(d/mn)$, highlighting a fundamental query–variance tradeoff (Liu et al., 2018a; Duchi et al., 2015). Yet, the simple averaging estimator in (1) may suffer from diminishing returns: increasing the number of queries per optimization step does not necessarily yield proportional variance reduction (Lin et al., 2025). This limitation motivates more principled strategies for allocating multiple perturbation queries, such as projection-alignment methods (Lin et al., 2025) that explicitly optimize the structure of the query matrix formed by multiple perturbations, or reusing queries across iterations (Wang et al., 2024; Qiu et al., 2025).

An extreme approach to variance reduction is to use a deterministic *coordinate-wise gradient estimator* (CGE), which achieves high-precision gradient estimation at a query cost that scales on the same order as the problem dimension, *i.e.*, $mn = d$. The CGE is given by (Liu et al., 2018b; 2020) $\hat{\nabla}_{\mathbf{x}} f(\mathbf{x}) = \sum_{i=1}^{d} \frac{f(\mathbf{x}+\mu\mathbf{e}_i)-f(\mathbf{x})}{\mu} \mathbf{e}_i$, where $\mathbf{e}_i$ denotes the $i$th canonical basis vector. Therefore, we argue that an effective query-variance tradeoff should achieve a *sublinear growth* in query cost with respect to the problem dimension $d$, thereby improving upon simple mini-batch strategies or CGE, whose query costs scale linearly with $d$. In contrast to (P1), (P2) emphasizes that pursuing the variance-reduced gradient estimation may incur prohibitively large query costs, which should be carefully weighed against the

constraints of the target application.

> **(P3) Do not forget the directional-derivative viewpoint.** The directional-derivative perspective provides a must-consider baseline for evaluating ZO optimization.

**Argument for (P3).** The finite difference $\frac{f(\mathbf{x}+\mu\mathbf{u})-f(\mathbf{x})}{\mu}$ in (RGE) converges to the *directional derivative (DD)* of $f$ at $\mathbf{x}$ along direction $\mathbf{u}$ as $\mu \to 0$ (Duchi et al., 2015). DD can be expressed as $f'(\mathbf{x}; \mathbf{u}) \triangleq \mathbf{u}^\top \nabla_\mathbf{x} f(\mathbf{x})$, with $\lim_{\mu \to 0} \frac{f(\mathbf{x}+\mu\mathbf{u})-f(\mathbf{x})}{\mu} = f'(\mathbf{x}; \mathbf{u})$. This connection suggests that the DD-based gradient estimator $f'(\mathbf{x}; \mathbf{u})\,\mathbf{u}$ should be treated as a must-consider baseline for ZO, against which RGE-type finite-difference estimators should be compared. This class of methods is commonly referred to as *forward gradient* or *directional gradient descent* (Baydin et al., 2022; Belouze, 2022; Ren et al., 2023; Silver et al., 2021). Furthermore, as we will show in Sec. 3, (P3) also provides a natural first lens for understanding and analyzing more sophisticated ZO gradient estimators.

**The absence of studies satisfying (P1)–(P3) simultaneously: An evidence from recent conferences.** To ground (P1)–(P3), we analyze some recent ICML'25, NeurIPS'25, and ICLR'26 papers on ZO optimization (**Table 1**). These works primarily develop ZO algorithms for two representative use cases: **(U1)** *large-scale model fine-tuning*, where ZO methods are adopted mainly for memory efficiency and deployment flexibility (Malladi et al., 2023; Yang et al., 2025); and **(U2)** *training from scratch*, where ZO optimization is applied without pretrained initialization due to inherently black-box model conditions (Chen et al., 2024).

*Table 1.* Summary of representative ZO optimization works and their alignment with positions (P1)–(P3).

| References | Use Cases | (P1) | (P2) | (P3) |
|---|---|---|---|---|
| ZO-NP (Sawada et al., 2025) | U2 | ✓ | ✗ | ✗ |
| AdaZO (Shu et al.) | U1 | ✓ | ✗ | ✗ |
| PaZO (Zhao et al., 2025a) | U1 | ✓ | ✗ | ✗ |
| Sparse MeZO (Liu et al., 2025b) | U1 | ✓ | ✗ | ✗ |
| PseuZO (Yue et al., 2025) | U1 & U2 | ✓ | ✗ | ✗ |
| SharpZO (Yang et al., 2025) | U1 | ✓ | Partially | ✗ |
| PAZO (Gong & Li, 2025) | U1 & U2 | ✓ | ✗ | ✗ |
| FZOO (Dang et al., 2026) | U1 | ✗ | ✓ | ✗ |
| OPZO (Xiao et al., 2026) | U2 | ✓ | ✗ | ✗ |
| HiSo (Li et al., 2026) | U1 | ✓ | ✗ | ✗ |

As shown in Table 1, (P1) has been the primary focus of existing work. This emphasis is unsurprising, as variance reduction is a key driver for improving ZO optimization performance. However, we argue that position (P2) should not be overlooked, particularly in the U2 setting, where query efficiency becomes a fundamental scalability bottleneck. Furthermore, (P3) is largely absent from recent works for both U1 and U2. Because the forward-gradient baseline exploits richer directional information than standard query-based ZO estimators, it provides a rough upper bound on the

performance that vanilla ZO optimization can potentially achieve. Moreover, it helps disentangle task difficulty from intrinsic ZO limitations: if the forward gradient succeeds while a ZO method fails, the bottleneck likely lies in the ZO method rather than in the task itself. To the best of our knowledge, the only existing work that includes (P3) is ZO-Bench (Zhang et al., 2024c), which compares forward-gradient methods against ZO methods.

## 3. Attention Required: Recommendations for Advancing ZO Optimization

In this section, we sharpen our positions (P4–P6) towards enabling technological breakthroughs in ZO optimization. These positions synthesize our perspectives with existing evidence, highlighting several high-potential directions that have thus far received relatively limited attention.

### 3.1. Subspace ZO optimization: Interpretable variance reduction with graceful query scaling

As discussed in (P2), variance reduction with a favorable query cost in ZO gradient estimation is crucial in improving ZO optimization performance, yet remains challenging in practice because estimator variance typically scales with the dimension $d$ of the optimization variables. This challenge naturally motivates a key question: *Can ZO gradient estimation be performed in a low-dimensional subspace of dimension $r \ll d$, such that the estimation variance is substantially reduced while maintaining a favorable query-variance tradeoff?* This is analogous to accelerating and stabilizing generative models (*e.g.*, diffusion models) by operating in a latent space rather than the original high-dimensional space (Rombach et al., 2022).

To address this challenge, several recent works have explored integrating *subspace optimization* into ZO methods (Chen et al., 2025; Yu et al., 2025; Nozawa et al., 2025; Gong & Li, 2025; Kozak et al., 2023; Lang et al., 2026a). These approaches reduce gradient estimation variance by introducing low-dimensional perturbation vectors (Yu et al., 2025; Nozawa et al., 2025; Gong & Li, 2025) or low-rank perturbation structures (Chen et al., 2025). This leads to the *subspace RGE (S-RGE)*, which modifies (RGE) to

$$\hat{\nabla}_\mathbf{x} f(\mathbf{x}) = \frac{f(\mathbf{x} + \mu\mathbf{P}\mathbf{u}) - f(\mathbf{x})}{\mu}\,\mathbf{P}\mathbf{u}. \qquad \text{(S-RGE)}$$

Here, $\mathbf{P} \in \mathbb{R}^{d \times r}$ with $r \ll d$ serves as a subspace projection matrix that maps *low-dimensional perturbations $\mathbf{u} \in \mathbb{R}^r$*, drawn from a standard Gaussian distribution, to full-space perturbations, thereby significantly reducing the number of perturbed coordinates. This is in contrast to (RGE) in which the original $\mathbf{u}$ is sampled in the full $d$-dimensional space.

Building on (S-RGE), we now articulate our position (**P4**), explaining why subspace ZO optimization constitutes a particularly promising direction from a new perspective.

**(P4) Subspace projection and spectral view.** Viewed through the directional-derivative lens (P3), S-RGE recovers projected FO gradients and admits a projected approximation of the original optimization problem in a low-dimensional subspace, yielding variance reduction on the order of $r \ll d$. This perspective further elevates subspace spectral structure as a powerful guiding principle for designing more effective ZO optimization methods.

**Argument for (P4).** If we understand S-RGE from the DD (directional derivative) lens in (P3), then we obtain $\hat{\nabla}_{\mathbf{x}} f(\mathbf{x}) = f'(\mathbf{x}; \mathbf{Pu})\mathbf{Pu} = [(\mathbf{Pu})^\top \nabla_{\mathbf{x}} f(\mathbf{x})]\mathbf{Pu}$ as $\mu \to 0$, where recall that $f'(\mathbf{x}; \mathbf{Pu})$ denotes the DD along the direction $\mathbf{Pu}$. Taking expectation over $\mathbf{u}$, where $\mathbf{u} \sim \mathcal{N}(\mathbf{0}, \mathbf{I}_r)$, yields $\mathbb{E}_{\mathbf{u}}\left[\hat{\nabla}_{\mathbf{x}} f(\mathbf{x})\right] = \mathbf{PP}^\top \nabla_{\mathbf{x}} f(\mathbf{x})$. If $\mathbf{P}$ is chosen to have orthonormal columns (*i.e.*, $\mathbf{P}^\top \mathbf{P} = \mathbf{I}_r$), then the matrix $\hat{\mathbf{P}} \triangleq \mathbf{PP}^\top$ admits a clear geometric interpretation: $\hat{\mathbf{P}}$ is a projection matrix, which projects any vector onto the subspace spanned by the columns of $\mathbf{P}$. Under this interpretation, the expectation of S-RGE, given by $\mathbf{PP}^\top \nabla_{\mathbf{x}} f(\mathbf{x})$, corresponds to the projection of the FO gradient $\nabla_{\mathbf{x}} f(\mathbf{x})$ onto the subspace defined by $\mathbf{P}$. The same interpretation was provided in (Lang et al., 2026a). Notably, S-RGE introduces only a modest computational overhead, which primarily arises from constructing the projection matrix $\mathbf{P}$. Existing works have suggested that a random $\mathbf{P}$, obtained via QR decomposition of a Gaussian matrix, can be effective in practice (Yu et al., 2025; Lang et al., 2026a). Furthermore, $\mathbf{P}$ can be updated lazily (Chen et al., 2025; Yu et al., 2025; Lang et al., 2026a), rather than at every iteration, which further reduces the projection overhead.

Based on the above discussion, we can clearly identify both the advantage and the inherent limitation of (S-RGE). On the positive side, projecting gradients onto a low-dimensional subspace substantially reduces estimation difficulty and variance; on the negative side, it may compromise gradient fidelity by discarding components orthogonal to the selected subspace. *Nevertheless*, when the underlying gradients exhibit low-rank structure, as is often observed in deep model training (Zhao et al., 2024; Kaushik et al., 2025), this subspace approximation can stay effective. Therefore, the variance-reduction benefits using a smaller subspace dimension $r$ would significantly outweigh the loss in gradient fidelity, resulting in substantial practical gains.

The subspace optimization perspective naturally connects to spectral optimization methods, particularly for matrix-structured parameters. Spectral optimization can exploit low-rank structures through reduced spectral subspaces, for example via Muon (He et al., 2025; Jordan et al., 2024; Liu et al., 2025a), which leverages gradient orthogonalization. This connection suggests a promising direction: integrating spectral optimization into subspace ZO methods to combine the variance reduction advantages of subspace optimization

with the favorable convergence behavior of spectral optimization. Such a combination could further enhance the scalability of ZO optimization (Lang et al., 2026a).

## 3.2. Distributed ZO optimization: Simplicity drives scalability and privacy

Because ZO gradient estimation relies only on forward function queries, it is simple to implement and deploy. This simplicity endows ZO optimization with exceptional flexibility for scalable, distributed, and system-aware learning, especially under communication, memory, or hardware constraints. For example, the mini-batch RGE in (1) and CGE admit a *finite-sum* structure, allowing workers to compute local estimates independently, then aggregate.

Beyond the finite-sum advantage, we articulate position **(P5)** to highlight the substantial systems-level potential of ZO optimization in distributed learning and parallel computing.

**(P5) Simplicity of ZO optimization as a systems advantage.** Decomposing ZO gradient estimation into scalar directional derivatives and reusable random perturbations enables highly communication-efficient distributed training, and its forward-only simplicity further supports efficient intra-machine and pipeline parallelism.

**Argument for (P5).** In distributed or decentralized learning settings, an often overlooked advantage of ZO optimization lies in its *communication efficiency* across workers (Fang et al., 2022; Li et al., 2025; 2026). As shown by RGE or S-RGE, a ZO gradient estimate naturally decomposes into a scalar finite-difference term (corresponding to a directional derivative) and a random perturbation vector. This decomposition enables inter-worker communication to transmit only the scalar finite-difference value, substantially reducing communication overhead. This is because the perturbation vector can be locally reconstructed by the receiver using a shared random seed from the transmitter to generate identical perturbations without explicit communication (Zelikman et al., 2023; Malladi et al., 2023). Therefore, the communication efficiency of ZO optimization stems from two key properties: the *scalar-valued directional derivative* in gradient estimation and the ability of *seed reuse* to reconstruct perturbations across workers.

The above communication efficiency also extends to S-RGE discussed in (P4). In a distributed setting with multiple workers, the random projection matrix $\mathbf{P} \in \mathbb{R}^{d \times r}$ can be generated from a shared random seed, eliminating the need to explicitly communicate $\mathbf{P}$. Each worker $i$ independently samples a random vector $\mathbf{u}_i \sim \mathcal{N}(0, I_r)$ using a worker-specific seed. The resulting local gradient estimator is

$$\hat{\nabla}_{\mathbf{x}} f_i(\mathbf{x}) = \underbrace{\left( \frac{f_i(\mathbf{x} + \mu \mathbf{Pu}_i) - f_i(\mathbf{x})}{\mu} \right)}_{:= \Delta_i} \mathbf{Pu}_i = \Delta_i \cdot \mathbf{Pu}_i. \quad (2)$$

Instead of transmitting the gradient estimate vector $\hat{\nabla}_{\mathbf{x}} f_i(\mathbf{x})$, worker $i$ only needs to communicate the scalar $\Delta_i$ together with the seed used to reproduce $\mathbf{u}_i$. The central node can then reconstruct the corresponding projected gradient and perform aggregation using the shared projection matrix $\mathbf{P}$.

Even on a single machine, ZO gradient estimation can be efficiently parallelized due to its simplicity of implementation. For example, when *structured perturbations* are used in ZO gradient estimation as discussed in (P1), whether at the coordinate-, layer-, or block-wise level, the resulting finite-difference computations can exploit *feature reuse* (Chen et al., 2024): By partially perturbing the optimization variables, only a subset of activations changes during function evaluation, enabling substantial computational savings. This allows the forward pass to reuse intermediate features up to the perturbed layer and compute only the remaining portion of the network, rather than recomputing from the input layer. A related idea is adopted in FZOO (Dang et al., 2026), which decomposes the computation into unperturbed and perturbation-specific components.

Furthermore, ZO optimization could fundamentally reshape the efficiency of *pipeline parallelism* (PP), a cornerstone of modern distributed training for foundation models (Huang et al., 2019; Shoeybi et al., 2019; Narayanan et al., 2019). In FO training, PP suffers from severe "pipeline bubbles" caused by the tight coupling between forward and backward passes and their sequential dependencies across stages (Narayanan et al., 2019; Qi et al., 2023). By eliminating backpropagation, ZO breaks this coupling: gradient estimates are obtained immediately after forward evaluations, enabling a unidirectional schedule. Thus, ZO pipelines could achieve near-zero bubbles and higher communication efficiency, effectively turning training into an inference-like workload that better saturates hardware resources.

The above positions ZO optimization as an inference-like, system-friendly paradigm, opening new opportunities for co-designing optimization algorithms with distributed, parallel, and hardware-aware learning systems. Its simplicity and reduced infrastructure requirements allow greater tolerance for using low-memory or legacy GPUs. From an energy perspective, this enables training on clusters of low-power GPUs, reducing both hardware and operational costs compared to relying exclusively on high-end accelerators.

**A privacy implication.** The simplicity of ZO optimization also introduces greater *noise* during training than FO methods, as discussed in (P1)–(P3). Yet, this noise can be beneficial rather than detrimental in certain settings, *e.g.*, by naturally strengthening privacy guarantees in *federated learning*. Recent works (Zhang et al., 2024a; Tang et al., 2025; Liu et al., 2024; Zhang et al., 2024b; Gong & Li, 2025) have investigated private ZO optimization. However, most existing approaches ensure privacy by explicitly adding Gaussian noise to ZO updates, closely mirroring differentially private perturbed GD/SGD mechanisms rather than fully leveraging the intrinsic noise and structural properties of ZO gradient estimation itself. We argue that differential privacy (DP) (Abadi et al., 2016) is instead a natural systems-level advantage of ZO optimization. In contrast to FO-based DP training (Abadi et al., 2016; Subramani et al., 2021), which typically relies on additional Gaussian noise injection, ZO optimization is inherently stochastic by design through the multiplication of a scalar finite-difference estimator with a Gaussian perturbation vector (see RGE and S-RGE). As a result, ZO methods can be naturally integrated into DP fine-tuning pipelines with minimal additional overhead.

### 3.3. Obfuscated ZO optimization: Task alignment as implicit obfuscation

Since the emergence of memory-efficient fine-tuning via ZO optimization (Malladi et al., 2023), this setting has become the dominant application of ZO methods, as reflected by the strong emphasis on U1 in Table 1. However, achieving competitive performance with ZO optimization in this regime critically depends on a prerequisite known as *task alignment* (Zhang et al., 2024c). That is, the fine-tuning task must be aligned with the pre-training objective via task prompts (Malladi et al., 2023) to effectively leverage the pre-trained initialization. Notably, such alignment is not required for FO fine-tuning.

This distinction implies that the apparent success of ZO optimization may stem less from the intrinsic optimization capability of ZO methods and more from the reduced task complexity induced by task alignment. Consequently, task alignment can act as an implicit form of *obfuscation*, making it difficult to disentangle whether performance gains should be attributed to ZO optimization itself or to the simplification of the underlying learning problem. Accordingly, position **(P6)** calls for future studies to explicitly disentangle the effectiveness of ZO optimization from task-complexity reductions induced by task alignment.

> **(P6) De-obfuscation from task alignment.** Evaluations of ZO optimization should explicitly disentangle the intrinsic optimization power from task-alignment gains.

**Argument for (P6).** We strongly recommend that ZO optimization should be evaluated under both *with-task-alignment* and *without-task-alignment* settings. The latter provides a more faithful characterization of the intrinsic ZO optimization capability. Closing this gap will lead to a clearer understanding of the true drivers of ZO optimization performance and will improve its generalizability to broader application scenarios where downstream tasks cannot be aligned with the pre-training objective.

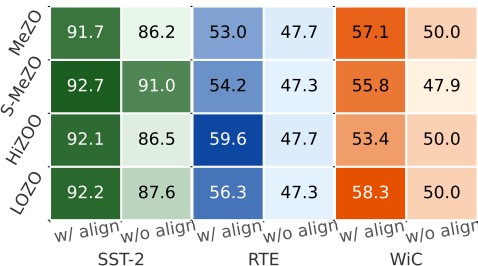

*Figure 2.* Fine-tuning accuracy of ZO optimization methods, including MeZO (Malladi et al., 2023), Sparse-MeZO (Liu et al., 2025b), HiZOO (Zhao et al., 2025b), and LOZO (Chen et al., 2025), on the SST-2, RTE, and WiC downstream tasks under task-aligned (*w/ align*) and non-aligned (*w/o align*) settings.

**Fig. 2** illustrates the fine-tuning accuracy of stateful ZO optimization methods, including MeZO (Malladi et al., 2023), Sparse-MeZO (Liu et al., 2025b), HiZOO (Zhao et al., 2025b), and LOZO (Chen et al., 2025), with and without task alignment when fine-tuning Gemma2-2B (Team et al., 2024) on downstream tasks SST-2, RTE, and WiC (Wang et al., 2019). Most ZO methods experience substantial accuracy drops without task alignment. Moreover, the relative ranking of methods changes between the aligned and unaligned settings, suggesting that task alignment can obscure the intrinsic capabilities of ZO optimization, which are more clearly disentangled when alignment is removed.

## 4. Call to Actions

Based on our positions in Secs. 2–3 (P1–P6), we next outline several concrete actions that the community can take to make meaningful progress in ZO optimization, moving beyond incremental algorithmic refinements.

**(A1) Establish rigorous, thorough, and de-obfuscated evaluation protocols for ZO optimization (in response to P2, P3, and P6).** We call for empirical studies to report performance, alongside computational and memory efficiency, under fixed query budgets or as a function of the query budget to expose the variance–query tradeoff; to explicitly incorporate forward-gradient baselines to calibrate the intrinsic capability of query-based ZO optimization; and to evaluate methods under both task-aligned and non-aligned settings to avoid obfuscated ZO optimization effects. Such protocols are essential for accurately characterizing the true strengths and limitations of ZO optimization, enabling fair comparisons across methods, and preventing over-attribution of performance gains to ZO algorithm design when they instead stem from favorable task structure.

**(A2) Move beyond full-space ZO optimization and rethink its relation to FO optimization (in response to P1, P2, and P4).** Many current obstacles in ZO optimization stem from operating in the full parameter space without exploiting underlying structure. Subspace projection and spectral optimization offer principled mechanisms for reducing variance while preserving the most informative optimization directions. Moreover, in settings where limited gradient information is available or can be obtained at low cost, hybrid FO–ZO approaches should be viewed not as violations of ZO purity, but as pragmatic designs that substantially improve the variance–query tradeoff. Treating ZO as part of a broader optimization continuum, rather than a strictly isolated paradigm, is essential for scalability (Gollapudi et al., 2026; Song et al., 2026). For example, from a game-theoretic perspective, ZO and FO can be seen as playing adversarial and defensive roles: ZO injects noise into descent directions, while FO denoises them. Under this view, alternating or hybrid ZO–FO optimization resembles adversarial training (Lang et al., 2026b) and may yield solutions with a flat minima (Zhang et al., 2026) and improved robustness. This also shares a similar spirit with sharpness-aware minimization (SAM) (Foret et al., 2021; Fan et al., 2025), which seeks flat minima during training by optimizing against worst-case model perturbations. In contrast, ZO optimization introduces random perturbations arising from the inherently biased ZO gradient estimator relative to the true FO gradient.

**(A3) Generative modeling for gradient estimation beyond classical ZO methods (in response to P1 and P2).** As a more ambitious direction beyond A2, we encourage the community to move beyond classical variance-reduction techniques and leverage learning-to-optimize approaches (Chen et al., 2022; Olsen et al., 2025) to develop *learnable* ZO optimizers that fundamentally rethink the process of gradient estimation itself. We highlight the promising role of generative models, especially *diffusion models* (DMs) (Yang et al., 2023), as learned gradient estimators. From a modeling perspective, ZO gradient estimates can be viewed as noisy observations of an underlying FO gradient, a structure that DMs are naturally suited to exploit through iterative denoising. Concretely, DMs could be trained to the model gradient distribution and thus to directly estimate FO gradients conditioned on RGE, potentially in a *ControlNet-style* conditional DM framework (Zhang et al., 2023).

**(A4) Build ZO-native system stacks (in response to P4).** We call for distributed frameworks explicitly designed for ZO optimization, rather than porting ZO algorithms into FO-centric systems such as DeepSpeed (Rasley et al., 2020), Megatron-LM (Shoeybi et al., 2019), or FSDP (Zhao et al., 2023a). Existing FO frameworks prioritize memory reduction at the cost of extra computation or communication (e.g., activation recomputation, tensor parallelism, or full sharding), trade-offs that are acceptable for FO training but *amplify ZO's computational burden and limit its scalability*.

We advocate a *ZO-native system design* that leverages ZO's

unique property: synchronization via lightweight scalars rather than full gradient tensors. Such frameworks should follow two principles. First, prioritize perturbation (query) parallelism over tensor parallelism: once a model replica is perturbed, it should be reused across as many micro-batches as possible, amortizing perturbation cost and reducing estimator variance. Second, exploit ZO's forward-only execution to enable dense pipeline scheduling. Unlike FO's 1F1B schedules (Narayanan et al., 2019), which suffer from bubble overhead, ZO-native pipelines can decouple stages to achieve near-zero bubbles and high hardware utilization. Only by shedding FO-specific constraints can ZO optimization realize its full potential at scale. With ZO-native infrastructure in place, many of the theoretical and algorithmic advantages of ZO optimization can be realized in practice, enabling large-scale models to be trained on collections of low-end GPUs in resource-limited environments by avoiding the memory and communication overheads of BP.

Another direction is to design *architectures* that are explicitly amenable to ZO optimization, including the joint exploration of model architectures and ZO strategies via neural architecture search and other automated design tools.

**(A5) Broaden the application frontier of ZO optimization.** We encourage the community to broaden the application scope of ZO optimization beyond current, often myopic use cases in deep learning.

One transformative opportunity lies in leveraging the *inference acceleration stack*. Unlike FO training, ZO optimization is dominated by forward evaluations, making its workload profile essentially identical to the rollout or serving phase in reinforcement learning (RL). This structural alignment allows ZO to directly benefit from advances in high-performance inference systems. We argue that future ZO solvers should be built atop inference engines (*e.g.*, vLLM (Kwon et al., 2023)) rather than traditional training stacks, thereby inheriting optimizations such as PagedAttention (Kwon et al., 2023), FlashAttention (Dao et al., 2022; Dao, 2023; Shah et al., 2024), and activation-aware quantization (Lin et al., 2024; Frantar et al., 2022). These inference optimizations naturally translate into training acceleration, capturing ZO gains that estimator-centric analyses miss.

In addition, *quantum and hybrid quantum–classical optimization* constitutes a particularly natural application domain for ZO methods (Arute et al., 2019; Gill et al., 2025; Kim et al., 2025). In quantum circuits, obtaining reliable FO gradients is fundamentally challenging: BP-style differentiation is in general incompatible with quantum measurements due to state collapse (Abbas et al., 2023), and hybrid quantum–classical training often suffers from vanishing gradients as system size grows (the *barren plateau* phenomenon) (Larocca et al., 2025). In contrast, ZO optimization aligns naturally with the measurement-based access

model in the quantum computing paradigm. Moreover, ZO gradient estimation can be performed exponentially faster on quantum devices (Jordan, 2005; Gilyén et al., 2019), and recent quantum convex optimization methods relying only on function queries can even match the query complexity of classical FO methods (Kim et al., 2025). These results suggest that ZO optimization is a *native* paradigm for quantum systems and motivate further development.

## 5. Alternative Views

Our position that ZO optimization is *underexplored rather than underpowered* contrasts with several perspectives in the literature. We discuss these alternative views below and clarify how our positions lead to different conclusions.

**Alternative view 1: ZO optimization is already sufficiently powerful in deep learning.** One view holds that ZO optimization is already a mature and effective paradigm. This belief is supported by two lines of evidence. First, ZO methods have achieved notable success in black-box, input-level applications such as adversarial example generation (Chen et al., 2017; Ilyas et al., 2018) and jailbreaking or promoting designs (Sun et al., 2022), where gradients are unavailable and model access is restricted to queries. Second, optimization theory establishes convergence guarantees for ZO optimization by interpreting finite-difference estimators as unbiased gradients of a smoothed objective (Duchi et al., 2015; Nesterov & Spokoiny, 2017), suggesting that ZO optimization admits guarantees comparable to those of FO methods under standard assumptions.

We argue that these evidences do not establish that ZO optimization is already sufficiently powerful for deep learning. Input-level applications operate in low-dimensional spaces, where the variance and query-scaling issues (highlighted in P1 and P2) are largely suppressed. Likewise, classical convergence guarantees, while important, are largely silent about the *resource constraints*, such as query budgets, memory, communication, and system efficiency, that govern practical performance in deep learning.

**Alternative view 2: ZO optimization is fundamentally limited and unlikely to scale.** A contrasting view argues that ZO optimization is intrinsically constrained by estimator variance and prohibitive query cost, rendering it unsuitable for large-scale deep models beyond niche or highly structured applications. This perspective is grounded in classical analyses showing that the variance of standard ZO gradient estimators scales linearly with the problem dimension, leading to poor sample and query efficiency in high-dimensional settings (Ghadimi & Lan, 2013; Ghadimi et al., 2016; Liu et al., 2018b).

We acknowledge that variance explosion and unfavorable variance–query tradeoffs pose real feasibility challenges for

ZO optimization. However, we argue that these difficulties stem primarily from how ZO methods are commonly instantiated, most notably via full-dimensional, element-wise gradient estimation and narrowly estimator-centric evaluations, rather than from fundamental limitations of ZO optimization itself. We suggest in (P4–P5) that ZO methods can instead operate in subspaces and unlock systems-level advantages largely absent from FO training. When these structural and systems properties are fully exploited, many perceived scalability barriers become far less severe.

**Alternative view 3: Continued progress hinges primarily on better gradient estimators.** Another influential perspective holds that advancing ZO optimization primarily reduces to designing increasingly sophisticated gradient estimators with improved statistical efficiency. This includes developing estimators with lower variance through batching, normalization, or structured perturbations (Liu et al., 2018a; Dang et al., 2026; Lin et al., 2025), and leveraging preconditioning or adaptive scaling strategies inspired by FO optimization (Zhao et al., 2025b;a).

While estimator design is undeniably important, an estimator-centric view is incomplete. As highlighted in (P5), ZO performance is tightly coupled with system execution models, such as distributed query parallelism, feature reuse, and pipeline scheduling, which improved estimators alone cannot address. Unlocking the full potential of ZO optimization also requires broadening the design space beyond the full element-wise space (P4) and rigorous, de-obfuscated evaluation protocols (P3, P6).

## 6. Related Work

In earlier sections, we have covered a broad range of existing ZO optimization work across the theory–algorithm– application stack. Here, we therefore focus on *nonstandard* ZO optimization approaches that still fall within the paradigm of gradient-free deep learning.

Distinct from finite-difference-based ZO methods, one prominent line of work on *deep learning without backpropagation* designs architecture-specific local objectives and update rules that rely only on local computations rather than global BP (Hinton, 2022; Ren et al., 2023). This direction is also motivated by the pursuit of biologically plausible learning mechanisms for deep networks (Xiao et al., 2018; Nøkland, 2016; Nøkland & Eidnes, 2019; Refinetti et al., 2020; Launay et al., 2020; Boopathy & Fiete, 2022). However, these approaches typically exhibit poorer scalability than ZO optimization (Chen et al., 2024).

Besides gradient-free methods tailored for deep learning, more classical black-box optimization approaches include *direct search*, *model-based* methods, *population-based* heuristics, and *Bayesian optimization*. Direct search ex-

plores the search space via structured or heuristic trial steps, with examples such as coordinate search (Fermi, 1952) and pattern search (Torczon, 1991; Chiang et al., 2022). Model-based methods instead build local surrogate models and optimize them iteratively, *e.g.*, through trust-region frameworks (Conn et al., 2000). Population-based heuristics include particle swarm optimization (Kennedy & Eberhart, 1995; Vaz & Vicente, 2009) and genetic algorithms (Grefenstette, 1993; Deb et al., 2002). Bayesian optimization fits probabilistic Gaussian processes (Shahriari et al., 2015), but is typically limited by the high cost of accurate surrogates in high dimensions. While these classical methods are effective for low-dimensional problems, they generally do not scale to the parameter regimes of modern deep learning.

## 7. Conclusion

We argue that the prevailing pessimism toward ZO optimization in deep learning is largely misplaced. While variance explosion and unfavorable query complexity pose real challenges, we contend that many perceived limitations stem not from fundamental barriers, but from overlooking ZO's unique strengths and from myopic, full-space, estimator-centric designs. By articulating positions (P1–P6) across the algorithmic, systems, and evaluation stack, we show that ZO optimization admits a much richer design space. Our central message is that ZO's simplicity, modularity, and systems advantages are not yet fully exploited. When they are, many supposed scalability barriers become substantially less severe. This can transform ZO from a niche workaround into a practical paradigm for training and deploying modern AI systems in constrained, hybrid, and emerging environments.

## Acknowledgments

This work was supported in part by the National Science Foundation (NSF) CAREER Award #2338068.

## Broader Impact

This work argues for a systematic rethinking of zeroth-order (ZO) optimization in deep learning, highlighting its underexplored potential across the algorithmic, systems, and evaluation stack. If successful, the perspectives advocated here could have several societal and environmental impacts.

*Democratizing access to large-scale model training.* A central motivation of ZO optimization is its forward-only nature and substantially reduced memory footprint. By removing the need for backpropagation and large activation storage, ZO methods can enable training and adaptation of large models on resource-constrained hardware, including older or low-memory GPUs. This lowers the barrier to entry for academic labs, small companies, and researchers in regions

with limited compute access, contributing to a more equitable research ecosystem.

*Enabling learning in hybrid, non-differentiable, and scientific systems.* ZO optimization is uniquely positioned to support end-to-end learning in hybrid pipelines that include simulators, symbolic components, tools, or physical processes where gradients are unavailable or unreliable (*e.g.*, digital twins, physics-informed systems, or agentic workflows). Broadening the application frontier of ZO methods (as advocated in the paper) could accelerate progress in scientific discovery, engineering design, and decision-making systems where current gradient-based methods are difficult to deploy. If ZO-native systems are realized, they could reduce reliance on high-end accelerators and enable the use of energy-efficient clusters, contributing to greener AI development.

*Privacy and robustness considerations.* ZO methods inherently inject noise through randomized perturbations and finite-difference estimation. While this noise is typically viewed as a drawback, it can also be a potential asset for privacy-preserving learning, especially in federated or distributed settings.

*Scientific impact and community norms.* This work advocates for changes in how the community evaluates and reports results (*e.g.*, fixed query budgets, forward-gradient baselines, and task-alignment de-obfuscation). These practices can improve reproducibility, transparency, and scientific rigor, reducing the risk of overclaiming progress due to confounded experimental setups.

*Risks and dual-use considerations.* Like any optimization technology, advances in ZO optimization could also be used to more efficiently adapt or optimize harmful models, including in black-box or restricted-access scenarios. For example, improved query efficiency could lower the cost of certain black-box attacks. This reinforces the importance of pairing algorithmic progress with responsible evaluation, access control, and monitoring mechanisms, and of continuing research on defenses, auditing, and governance of powerful models.

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
