# OpenReview forum: "Position: Zeroth-Order Optimization in Deep Learning Is Underexplored, Not Underpowered"
_ICML.cc/2026/Position_Paper_Track — ICML 2026 Position Paper Track spotlight_

### Official Review · Reviewer_Kytw · 2026-03-09

**Significance:** 4
**Argument Clarity:** 3
**Rating:** 6
**Confidence:** 5

**Questions:**

1.Do the authors consider increasing ZO optimization work in journals and more conferences in Table 1 to support their arguments?

2.Is there no work using directional derivative to evaluate its ZO method?

3.How to accurately define and quantify the degree of 'task alignment'?

**Alternative Views Section:**

Yes

**Compliance With Llm Reviewing Policy A Conservative:**

Affirmed.

**Discussion Potential:**

4

**Final Justification:**

The rebuttal addressed my main concerns,  and I greatly agree with the acceptance of this paper.

**Paper Summary:**

The core argument of this paper, "Zeroth-order optimization (ZO) is underexplored, not underpowered," is clear and powerful. It raises systematic questions and responds to the fueled skepticism towards ZO methods in the current deep learning community. The authors construct six positioning arguments (P1-P6) spanning three dimensions: algorithm, system, and evaluation, and propose concrete actions (A1-A5). Not only does it point out the limitations of existing research, but it also calls for promising future directions.

**Position:**

Yes

**Position In Title:**

Yes

**Related Work:**

3

**Strengths And Weaknesses:**

Strengths:

1.The primary contribution of this paper lies in uncovering the issues (such as mainly centering around estimators) of current ZO research rather than merely improving individual ZO algorithms. It proposes a comprehensive research framework encompassing algorithmic foundation (P1, P2, P4), infrastructure (P5), and evaluation (P3, P6), which provides a clear roadmap for future research.

2.P1-P3 solidly reviewed the fundamental issues often overlooked in ZO research, such as random direction selection, query variance trade-off, and directional derivative baseline. The analysis of NeurIPS'25 work (Table 1) effectively supports the argument that "current research is not comprehensive". Especially for P3, as the authors mentioned, it helps disentangle task difficulty from intrinsic ZO limitations: if the forward gradient succeeds while a ZO method fails, the bottleneck likely lies in the ZO method rather than in the task itself.

3.P4 (Subspace projection and spectral view) and P5 (System advantage) indicate the key paths to breaking through the current bottleneck. For example, the combination of ZO with subspace/spectral methods can substantially reduce the estimation variance while maintaining a favorable query-variance tradeoff (P2). The simplicity and finite-sum structure of ZO optimization show exceptional flexibility for scalable, distributed, and system-aware learning, especially under communication, memory, or hardware constraints, when decomposing ZO gradient estimation into scalar directional derivatives and reusable random perturbations.

4.P6 (Task alignment) uncovers the additional requirement of ZO optimization over FO optimization that the fine-tuning task must be aligned with the pre-training objective, which calls for future studies to explicitly disentangle the effectiveness of ZO optimization from task-complexity reductions induced by task alignment.

5.This paper proposes some useful concrete actions, e.g., (1) establishing rigorous, thorough, and de-obfuscated evaluation protocols for ZO optimization; (2) moving from full-space to sub-space; (3) combining with FO optimization; (4) replacing classical ZO methods with learnable ZO optimizers, e.g., generative modeling; (5) building ZO-native system stacks; (6) broadening the application of ZO, e.g., inference acceleration stack.

6.This paper discusses several alternative views to support its argument, i.e., (1) ZO optimization is already sufficiently powerful; (2) ZO optimization is fundamentally limited and unlikely to scale; (3) Continued progress hinges primarily on better gradient estimators.

Weaknesses:

1.In the right-hand side of line 113-114, there should be “Here $u_i$ and $\epsilon_j$ denote the i-th random vector and j-th sample”.

2.This paper discusses the subspace method (P4) and system advantages (P5) separately, but there is a strong potential for collaboration between P4 and P5. For example, how does subspace projection affect communication patterns in distributed training? Can low rank perturbation structures bring new designs to 'query parallelism'? It is suggested to briefly explore the new research directions that may arise from the cross-fusion of P4 and P5.

3.It is recommended to double-check whether there are any latest relevant preprints from the end of 2025 or early 2026 that can be included in the discussion to maintain their timeliness.

**Support:**

3

---

> ### Author Rebuttal · Authors · 2026-03-30
>
> We thank the reviewer for the insightful comments. Please find our detailed responses to the questions below.
>
>
> **Q1. Typo.**
>
> **A1.** Thanks, we will correct this typo in our revision.
>
> **Q2. Potential for collaboration between P4 and P5. For example, how does subspace projection affect communication patterns in distributed training? Can low rank perturbation structures bring new designs to 'query parallelism'?**
>
> **A2.** Good point! Subspace-ZO (P4) preserves and can further strengthen the system advantages in P5. In a distributed setting with $W$ workers, the projection matrix $P \in \mathbb{R}^{d \times r}$ can be generated from a shared random seed, so $P$ itself does not need to be communicated. Each worker $w$ samples a unique $u_w \sim \mathcal{N}(0, I_r)$ using a worker-specific seed. The local gradient estimate is:
> $$\hat{\nabla}_{x} f_w(x) = \left( \frac{f_w(x + \mu P u_w) - f_w(x)}{\mu} \right) P u_w = \Delta_w \cdot P u_w$$
>
> Instead of transmitting the full $d$-dimensional vector $\hat{\nabla}_{x} f_w(x)$, worker $w$ only sends the scalar $\Delta_w$ along with the seed used to generate $u_w$. The central node then reconstructs and aggregates the gradients using the shared $P$.
>
> This shows that subspace structure allows low-rank, communication-efficient query parallelism, directly connecting P4 and P5. We will add this discussion in the revision.
>
> **Q3. Latest relevant preprints that can be included in the discussion**
>
> **A3.** Thanks for the insightful comment. We have identified additional works (including recent ones published after the ICML submission deadline) that further support our positions.
>
> **P1 & P4:** Structured queries, subspace and spectral view: ZO-Muon [1] explicitly unifies a subspace view with Muon-style spectral optimization. AGZO [2] proposes extracting a compact, activation-informed subspace on the fly during the forward pass and restricting perturbations to this low-rank space. LOREN [3] moves beyond gaussian perturbations by capturing anisotropic local curvature using a low-rank block diagonal preconditioner to dynamically learn a perturbation distribution.
>
> **P5:** System-level advantage: [4] Introduces a federated optimization method that strictly preserves scalar-only communication while leveraging global diagonal Hessian approximations to accelerate convergence. ZOWarmUp [5] Leverages the communication efficiency of ZO methods to enable training from random initialization across heterogeneous edge devices.
>
> [1] Lang, et al. "Powering Up Zeroth-Order Training via Subspace Gradient Orthogonalization."
>
> [2] Lin, et al. "AGZO: Activation-Guided Zeroth-Order Optimization for LLM Fine-Tuning."
>
> [3] Seung, et al. "Low-Rank Curvature for Zeroth-Order Optimization in LLM Fine-Tuning."
>
> [4] Li, et al. "Converge Faster, Talk Less: Hessian-Informed Federated Zeroth-Order Optimization." The Fourteenth International Conference on Learning Representations.
>
> [5] Legate, et al. "Warming Up for Zeroth-Order Federated Pre-Training with Low Resource Clients."
>
> **Q4. Increasing ZO optimization work in journals and more conferences in Table 1 ?**
>
> **A4.** We thank the reviewer for the constructive feedback. We have sourced ICLR 2026 and ICML 2025 papers to support our argument, shown in the table below. We are also open to including relevant journal works.
>
> | References | Use Case | P1 | P2 | P3 |
> | :--- | :--- | :---: | :---: | :---: |
> | ZO-NP [1] | U2 | ✓ | ✗ | ✗ |
> | AdaZO [2] | U1 | ✓ | ✗ | ✗ |
> | FZOO [3] | U1 | ✗ | ✓ | ✗ |
> | OPZO [4] | U2 | ✓ | ✗ | ✗ |
> | HiSo [5] | U1 | ✓ | ✗ | ✗ |
>
> [1] Sawada, et al. "Natural perturbations for black-box training of neural networks by zeroth-order optimization."
>
> [2] Shu, et al. "Refining Adaptive Zeroth-Order Optimization at Ease."
>
> [3] Dang, et al. "FZOO: Fast Zeroth-Order Optimizer for Fine-Tuning Large Language Models towards Adam-Scale Speed."
>
> [4] Xiao, et al. "Online pseudo-zeroth-order training of neuromorphic spiking neural networks."
>
> [5] Li, et al. "Converge Faster, Talk Less: Hessian-Informed Federated Zeroth-Order Optimization."
>
>
> **Q5. Is there no work using directional derivative to evaluate its ZO method?**
>
> **A5.**  As shown in **Table 1** in our submission, directional derivative is largely overlooked in recent ZO literature, with few exceptions [1]. In the broader context of BP-free training, directional-derivative methods are known as forward gradients [2–3], yet they are rarely compared with ZO approaches. This gap motivates our advocacy for incorporating the directional-derivative perspective in ZO training.
>
> [1] Zhang, et al. "Revisiting Zeroth-Order Optimization for Memory-Efficient LLM Fine-Tuning: A Benchmark."
>
> [2] Baydin, et al. "Gradients without backpropagation."
>
> [3] Ren, et al. "Scaling Forward Gradient With Local Losses."

---

> > ### Author Rebuttal · Reviewer_Kytw · 2026-04-02
> >
> > Thanks for the detailed response. However, Q3 remains unanswered. I am not clear about the details of the comparison in Figure 2. Besides, I want to know if there is any work using de-obfuscation from task alignment to evaluate its ZO method.

---

### Official Review · Reviewer_A87i · 2026-03-09

**Significance:** 3
**Argument Clarity:** 3
**Rating:** 5
**Confidence:** 2

**Questions:**

1. Are there empirical studies demonstrating that ZO methods  scale effectively to large modern models (eg LLM fine-tuning) compared to existing approaches?

2. Several future directions (eg, diffusion models for gradient estimation) are proposed. Do the authors have preliminary experiments or proof-of-concept results supporting the feasibility of these ideas?

**Alternative Views Section:**

Yes

**Compliance With Llm Reviewing Policy A Conservative:**

Affirmed.

**Discussion Potential:**

4

**Final Justification:**

I support acceptance of the paper.

**Paper Summary:**

The paper argues that zeroth-order optimization in deep learning is underexplored rather than fundamentally limited, and proposes 6 positions spanning algorithmic design, systems considerations, and evaluation methodology to guide future research directions in ZO optimization.

**Position:**

Yes

**Position In Title:**

Yes

**Related Work:**

4

**Strengths And Weaknesses:**

**Strengths**
* The paper clearly states and defends the main claim that the perceived limitations of ZO optimization arise largely from current research practices. The six positions P1-P6 provide a structured framework that spans algorithmic considerations, systems aspects, and evaluation protocols, which makes the argument logical and easy to follow.
* With the recent interest in memory-efficient training and gradient-free learning, revisiting ZO optimization in the context of modern deep learning is both timely and relevant. The discussion connects ZO optimization to current research on LLM fine-tuning, distributed training, hybrid ML systems, and resource-constrained environments. This makes the position very interesting for the research community.
* The paper highlights several directions that are underexplored in the literature. These perspectives are likely to stimulate discussion and could help broaden the way the community evaluates ZO methods.

**Weaknesses**
* Despite extensive discussion and citing related work, the main arguments remain largely conceptual. Some key claims, for example regarding the scalability benefits of subspace ZO optimization or the systems advantages of forward-only execution, would benefit from stronger empirical evidence or clearer quantitative comparisons. Also see questions below.
* Several proposed directions, such as generative-model-based gradient estimation, are interesting but remain speculative. The paper would be stronger if it provided preliminary evidence, case studies, or clearer feasibility arguments to support such claims. Also see questions below.

**Support:**

3

---

> ### Author Rebuttal · Authors · 2026-03-30
>
> We thank the reviewer for the insightful comments. Please find our detailed responses to the questions below.
>
> **Q1. Despite extensive discussion and citing related work, the main arguments remain largely conceptual. Some key claims would benefit from stronger empirical evidence or clearer quantitative comparisons.**
>
> **A1.** We respectfully clarify that our submission is a **position paper**, whose primary objective is to provoke discussion, synthesize emerging optimization paradigms, and identify key bottlenecks. Similar to other position papers at ICML’25, e.g., the outstanding papers [1–2], it should not be penalized for the absence of extensive experiments. Importantly, our arguments are grounded in both direct and indirect evidence from the literature (as provided in our paper), and we also include targeted empirical results (e.g., for P6) to support our positions.
>
>
> [1] Hazra, Sanchaita, Bodhisattwa Prasad Majumder, and Tuhin Chakrabarty. "Position: AI Safety should prioritize the Future of Work." Forty-second International Conference on Machine Learning Position Paper Track. 2025.
>
> [2] Kim, Jaeho, Yunseok Lee, and Seulki Lee. "Position: The AI Conference Peer Review Crisis Demands Author Feedback and Reviewer Rewards." Forty-second International Conference on Machine Learning Position Paper Track. 2025
>
>
> **Q2. Several proposed directions, such as generative-model-based gradient estimation, are interesting but remain speculative. The paper would be stronger if it provided preliminary evidence, case studies, or clearer feasibility arguments to support such claims.**
>
> **A2.** To clarify, applying generative models for ZO estimation falls within the broader framework of **Learning to Optimize** (L2O). Prior work [1] shows that neural networks (without generative modeling) can learn ZO gradient estimators. We suggest conditional generative models as a potentially more powerful and flexible extension. That said, as noted in our response to Q1, this is a position paper aimed at synthesizing trends and highlighting underexplored directions. Direction A3 is intentionally framed as an ambitious but promising research direction, not a finalized method.
>
>
> [1] Zhang, et al. "Learning a Zeroth-Order Optimizer for Fine-Tuning LLMs."

---

> > ### Author Rebuttal · Reviewer_A87i · 2026-04-01
> >
> > I thank the authors for the rebuttal and clarifications.

---

### Official Review · Reviewer_Mc5x · 2026-03-16

**Significance:** 3
**Argument Clarity:** 4
**Rating:** 5
**Confidence:** 3

**Questions:**

I think this position paper is meaningful and sufficiently strong, I have not identified critical limitations and recommending "accept". For the decision process it would be helpful to know how the questions Q1-5 above can be clarified / addressed or if some other points are mistaken.

**Alternative Views Section:**

Yes

**Compliance With Llm Reviewing Policy A Conservative:**

Affirmed.

**Discussion Potential:**

3

**Final Justification:**

After the rebuttal and reading other reviews, my evaluation does not change, I recommend accepting the paper. I am sure the authors will take a note of all the discussions.

**Paper Summary:**

The paper, based on the recent advances in zero order optimization (ZO) and its common practices in a very systematic manner delineated limitations and underexplored directions. It makes the case that there are many perspectives for improving ZO efficiency e.g. by leveraging problem structure, for discovering benefitting applications, for optimizing the software to maximize its efficiency. The paper also discusses the (apparently poor) state of the art evaluation methodologies. It contains calls for action, many of which can be only achieved as a community effort, such as developing finer standards for evaluation, infrastructure, pursuing broader exploration.

**Position:**

Yes

**Position In Title:**

Yes

**Related Work:**

4

**Strengths And Weaknesses:**

This position paper is very concrete, well articulated, backed with SOTA literature trends, common practices and examples. It is very clear about several positions, backed by reasonable arguments. The topic already attracts high interest due to recent applications in memory-constrained fine-tuning and handling non-differentiable components. It addresses the community calling for a broader exploration and deeper evaluations, allowing to assess methods efficiency and delinitate it from other factors. Although, some points made would be rather obvious to experienced researchers (e.g. about taking the computation cost into account in comparisons), overall they seem to be important, motivating/inspiring and in some cases discussion-provoking.

One point that may seem obvious, but which I find actually really cool, is that of estimating the derivative on a subspace. From the perspective of ZO, estimating the derivative w.r.t. a subset of $r$ coordinates or the derivative projection to a $r$-dimensional subspace (which is the same) is trivially cheaper. So far the math is obvious: we need $2r$ forward passes to get $r$ "coordinates" of the gradient -- poor efficiency. However the paper hints at the following questions: do we need full space gradient *per data point* for fine-tuning? It is a common practice in fine-tuning to adjust a small subset of parameters (e.g. affine or biases only, adaptors, low-rank corrections). So probably not. Then, do we necessarily need it in expectation for full optimization? When subspaces are selected in the data and training-dependent manner, things get more interesting, e.g. for $f(w^\top x)$ the gradient is $f'(\cdot)  x$ and so it is reasonable to explore the direction $\Delta w \propto x$. In this case we actually use the structure of the chain rule and conclude that ``structure-aware'' ZO estimate is the same as FO gradient. So I don't know how far it gets with structure-aware directions in more general cases, and what optimization-related questions arise. Maybe it could be also related (and provide a rationell) to local learning rules based on activation correlations?

For the weaknesses I would like to bring up the following:
- **Q1** The use cases such as differentiable (U1) and non-differntiable (U2) are somewhat mixed together, i.e when suggesting to use directional derivative as a baseline.
- **Q2** Gradients exhibit low-rank structure (line 223) does not imply that we can use this structure for ZO: e.g. if we new the true gradient $g$ we could use the direction $u = g/\| g\|$ to measure, but we of course don't know it.
- **Q3** The adversarial ZO-FO seems to go towards something like Sharpness-Aware Minimization (SAM). But I think this is off-topic, not about relation of FO and ZO.
- **Q4** The diffusion model idea seemed to me too unclear / stretched. Because a generative model would need to be trained and for a specific parameter point. And it probably is not worth it just for getting a gradient estimate. It would be easier to go for meta-learning: just train a transformer that given a set of user data and network forward activations for them directly predicts the fully tuned model. E.g. Liang et al. Drag-and-drop LLMS: zero-shot prompt-to-weights, 2025.
- **Q5** I did not understand why properties of ZO optimization methods would be more clearly disentangled when  the alignment is removed. Is it method-dependent or dynamically interferes with the method? If yes, it is a part of method. If not, all methods can be applied with the same alignment -- in this case it changes the characteristics of the problem. But if it is a relevant part (leads to better improvement in practice), why is it necessary to remove it? Is it not like saying that e.g. "data standardization" or "He initialization" is obfuscating optimizers comparison and should be removed?

Let me also write some further thoughts --- a relevant feedback not necessarily reflecting weakness from the reviewing / paper acceptance perspective. I have a somewhat alternative view. This is how I would analyze different use cases, from a pragmatic perspective:

1. If the model is a white-box and differentiable.
The time of FW-BW is just twice as much as FW (even if the activation maps need not be stored for backward they are still written and red, so memory IO stays). It is about the same time as of computing just 1 gradient coordinate with the forward differentiation (ZO). So, if memory allows, no reason to use ZO. And if it is an LLM, the decoding needs to keep KV cache in memory anyhow, with that same memory parallel BW is feasible with maybe twice shorter context. Let's say it is some other model, e.g. a vision transformer and the memory is limited. Efficiency-wise, it is best to fine-tune it in the cloud. Well, if you really have to do it on the edge, there are still several options to cut down on the memory: checkpointing, quantizing saved activations (ActNN: Reducing Training Memory Footprint via 2-Bit Activation Compressed Training) and more. Actually, I can hardly think of a realistic scenario where backprop would be so intractable (staying in the case of white-box differentiable) and have such a high variance as to resort to ZO. At least, I think this more expanded version of the FO side and its tradeoffs need to be considered on the alternative side to ZO subspace techniques and their complexity. And then there are zero-shot solutions (Drag-and-drop LLMS mentioned above), with their, possible big, but non-iterative costs. From that perspective calling for optimizing *parallel white-box pipelines* for ZO, especially utilizing parallelism over queries (parameter perturbations), which would take proportionally more memory and compute, seems to me like promoting inefficient use of compute.
2. If the model is a white-box but not differentiable. Here we could consider some alternatives, exploiting the known structure as well as possible. For instance for combinatorial solvers we could use Blackbox Backpropagation (Differentiation of Blackbox Combinatorial Solvers). Or, if it is a network with quantized activations, we could use REINFORCE (possibly with several samples) for the expected loss with noisy stochastic activatinos. Same for the case when weights are quantized. If it is something like ODE it is differentiable through other means than backprop.
3. If the model is a black-box. First of all, if it is a black box merely for the reason that you do not have access to it -- do not do it all (I mean the research community should not address such adversarial problems). Second of all, ZO cannot take advantage of the structure to construct directions (because the function is blackbox) and software pipeline cannot be optimized for ZO, to reuse activatinos, etc. (again, because the function is a blackbox!). So perhaps, I have misunderstood something, but it seems that ZO stays principally limited and underpowered here. Using surrogate models of a kind appear more promising.

**Support:**

3

---

> ### Author Rebuttal · Authors · 2026-03-30
>
> We thank the reviewer for the insightful comments. Please find our detailed responses to the questions below.
>
> **Q1. The use cases such as differentiable (U1) and non-differntiable (U2) are mixed together when suggesting to use directional derivative as a baseline.**
>
> **A1.** To clarify, U2 is not inherently non-differentiable; prior work [1] applies ZO to simulation-coupled deep models that remain differentiable. Furthermore, in non-differentiable settings, randomized smoothing (Line 123-124) can make the objective differentiable [2],  allowing directional derivative to be used.
>
> [1] Chen, et al. "DeepZero: Scaling Up Zeroth-Order Optimization for Deep Model Training."
>
> [2] Duchi et al. "Randomized Smoothing for Low-Complexity Optimization."
>
>
> **Q2. Gradients exhibit low-rank structure does not imply that we can use this structure for ZO.**
>
> **A2.** Similar to FO subspace methods (e.g., LoRA), the projection rank in S-RGE is a tunable hyperparameter, not something reliant on the ground-truth gradient. Empirically, moderate ranks (e.g., $r=64$) in prior ZO work [1] capture meaningful low-dimensional directions and significantly reduce estimation variance without needing to match the true rank.
>
> [1] Yu, et al. "Zeroth-order fine-tuning of llms in random subspaces."
>
> **Q3. The adversarial ZO-FO seems to go towards something like SAM.**
>
> **A3.** We agree there is conceptual overlap, as both use weight perturbations to improve smoothness or noise-tolerance. We will clarify the key distinction in our revision: SAM uses adversarial perturbations to seek flat minima, whereas ZO relies on random perturbations for gradient estimation.
>
> **Q4. The diffusion model idea seemed unclear/stretched. It would be easier to go for meta-learning.**
>
> **A4.** Great point for discussion. First, applying generative models for ZO estimation falls within the broader framework of Learning to Optimize (L2O). Prior work [1] has shown that neural networks (without generative modeling) can learn ZO gradient estimators. We suggest (conditional) generative models as a potential extension with greater flexibility. Note that meta-learning is also related to L2O and is not necessarily simpler.
>
> [1] Zhang, et al. "Learning a Zeroth-Order Optimizer for Fine-Tuning LLMs."
>
> **Q5. Why properties of ZO would be more clearly disentangled when the alignment is removed.**
>
> **A5.** Great question! Our point is about evaluation, not deployment: to assess a ZO optimizer, we should isolate its effect from changes in task difficulty.
>
> We view task alignment as fundamentally different from techniques like data standardization or initialization. The latter condition the optimization landscape, while alignment can alter the task itself (e.g., by making it closer to the pre-training objective), thereby reducing intrinsic difficulty.
> Evaluating without alignment is useful for two reasons: (1) Fair comparison to FO methods, which do not rely on alignment; (2) Avoiding confounding effects, as alignment can change the relative ranking of methods (as shown in **Fig. 2**), obscuring true ZO optimization capability.
>
> We do not suggest removing alignment in practice; rather, we advocate evaluating ZO methods both with and without alignment to obtain a clearer assessment.
>
> **Q6. Alternative views when the model is a white-box and differentiable.**
>
> **A6.** While FO is preferred under unconstrained memory and communication, this rarely holds in practice:
> * **Communication & pipeline efficiency:** FO requires $O(d)$ communication per step; ZO requires $O(1)$ via shared seeds. Being backward-free, ZO pipeline stages compute gradients locally and simultaneously, yielding near-zero pipeline bubbles (lines 241-254).
> * **Strict memory limits:** FO memory scales poorly with sequence length due to gradients and optimizer states. ZO eliminates backpropagation, strictly bounding memory to inference levels. This enables long-context training where FO simply goes out of memory [1].
>
> [1] Zhang, et al. "Revisiting Zeroth-Order Optimization for Memory-Efficient LLM Fine-Tuning: A Benchmark."
>
> **Q7. Alternative views when the model is a white-box but not differentiable.**
>
> **A7.** We agree. However, ZO remains applicable here since it relies on randomized smoothing [1]. This provides unbiased gradients of a smoothed objective  (Line 123-124), which are differentiable regardless of the original objective's differentiability.
>
> [1] Duchi, et al. "Optimal rates for zero-order convex optimization: The power of two function evaluations."
>
> **Q8. Alternative views when the model is a black-box.**
>
> **A8.**  Black-box settings extend far beyond adversarial ML. In many scientific and engineering contexts (e.g., digital twins, solver-in-the-loop), the objective involves non-differentiable simulators lacking internal gradients. ZO has proven highly effective in these structurally complex settings [1].
>
> [1] Chen, et al. "DeepZero: Scaling Up Zeroth-Order Optimization for Deep Model Training."

---

> > ### Author Rebuttal · Reviewer_Mc5x · 2026-04-03
> >
> > I thank the authors for detailed clarifications and the references.
> >
> > A quick follow-up question:
> > For LLM inference (decoding) one needs to have the memory that can hold all past key-value pairs for all layers. Why do you say that FO goes out of memory on the same HW? Does it take more memory? - yes, but by a small constant factor. It scales no more poorly than the inference itself. Does it need to run with the maximum inference context length? -- I think no.

---

### Official Review · Reviewer_ouyd · 2026-03-17

**Significance:** 4
**Argument Clarity:** 3
**Rating:** 5
**Confidence:** 3

**Questions:**

See weaknesses

**Alternative Views Section:**

Yes

**Compliance With Llm Reviewing Policy A Conservative:**

Affirmed.

**Discussion Potential:**

4

**Final Justification:**

I thought it was a good paper, and I was satisfied with the rebuttal.

**Paper Summary:**

This paper argues against the prevailing skepticism about using zeroth-order optimization techniques for Neural Networks. They argue that this direction underexplored rather than underpowered and improving development practices can make them competitive. They make the argument for this by dividing their position into 6 subpositions:
1. Perturbation directions matter for variance of the gradient estimation.
2. There is a tradeoff between query complexity and variance
3. Directional derivative should be used as a baseline
4. Subspace spectral structure as a powerful guiding principle for designing more effective ZO optimization methods
5. Simplicity of ZO makes it especially suitable in distributed or decentralized learning settings
6. Evaluations should disentangle ZO optimization power from task alignment.
They provide arguments for each subposition and outline the future research directions

**Position:**

Yes

**Position In Title:**

Yes

**Related Work:**

4

**Strengths And Weaknesses:**

# Strengths
1. This is a very well-argued paper for an ambitious and wide position. Due to the generality of the position as stated in the title, dividing it into more concrete arguments was a good choice.  P2 and P6 are good examples of this, where they suggest very concrete research directions.
2. This paper cites very widely and they generally do a good job of supporting the arguments in the paper

# Weaknesses
1. I found the discussion around P5 to be a bit unclear. I am unsure why ZO methods would give us an advantage over FO methods. Isn't the cost  (and parallelizability) of the backward pass the same as a forward pass. I am not able to see the relevant difference in these two settings for either privacy or scalability
2. I think P4 would benefit from a discussion about the cost of what seems to be fairly expensive linear algebra operations.

**Support:**

4

---

> ### Author Rebuttal · Authors · 2026-03-30
>
> We thank the reviewer for the insightful comments. Please find our detailed responses to the questions below.
>
> **Q1.  Found the discussion around P5 to be unclear. Unsure why ZO methods would give us an advantage over FO methods.**
>
> **A1.** We thank the reviewer for raising this question. To clarify, the primary system-level advantages of ZO optimization discussed in **P5** do not stem from raw FLOP reductions, but rather from its ability to alleviate communication bottlenecks and simplify pipeline structures in distributed training.
>
> In modern large-scale distributed training, communication, not computation, is typically the primary bottleneck [1]. By dramatically reducing communication volume and simplifying pipeline structures, ZO achieves greater wall-clock efficiency than FO optimization, even if FLOPs are comparable. Specifically, ZO offers advantages in four key areas:
> * **Communication efficiency:** FO distributed training requires all-reduce ($O(d)$ communication) per iteration to sync the gradients from different ranks, which are of the same size of model parameters. In contrast, ZO only needs to share random seeds used to generate the perturbations and broadcast a single scalar (finite-difference value) to reproduce the “estimated gradient” in other ranks ($O(1)$ communication), drastically reducing network overhead. See  lines 262-274 in our submission.
> * **Parallelizability (zero bubble):** FO pipeline parallelism (PP) suffers from idle "bubbles" due to the strict sequential dependency between forward and backward passes. Because ZO is entirely backward-free, all pipeline stages could compute their gradient estimates locally and simultaneously, possibly achieving near-zero pipeline bubbles (see lines 241-254).
> * **Wall-clock time vs. FLOPs:** Relying solely on FLOPs does not reflect true training efficiency. While an FO backward pass is theoretically 2x the FLOPs of a forward pass [2], modern LLM training is primarily bottlenecked by memory and data movement, not compute [1]. ZO entirely eliminates the need to store intermediate activations (reducing memory footprint by up to 10x [3], which is proportional to batch size and context length). Consequently, ZO can achieve significantly shorter wall-clock times in distributed setups.
> * **Privacy for “free”:**   Privacy-aware FO methods typically require explicit Gaussian noise injection to achieve differential privacy (DP), however, ZO gradient is intrinsically stochastic by design (scalar finite-difference multiplying a Gaussian vector) and can be seamlessly integrated into DP fine-tuning without extra overhead (lines 266-278).
>
> [1] Ivanov, Andrei, et al. "Data movement is all you need: A case study on optimizing transformers."
>
> [2] Kaplan, Jared, et al. "Scaling laws for neural language models."
>
> [3] Malladi, Sadhika, et al. "Fine-tuning language models with just forward passes."
>
>
> **Q2. I think P4 would benefit from a discussion about the cost of what seems to be fairly expensive linear algebra operations.**
>
> **A2.** Great question! We will include a more detailed discussion of computational costs.
>
> We clarify that the overhead of Subspace-ZO (**S-RGE**) is modest and mainly comes from constructing the projection matrix. In a matrix-wise implementation, for each parameter $X \in \mathbb{R}^{m \times n}$, a projection matrix $P \in \mathbb{R}^{m \times r}$ ($r \ll m,n$) is obtained via QR decomposition, incurring a one-time cost of $O(mr^2)$. However, in practice, $P$ can be updated lazily (e.g., every $V=100$ steps [1-2]), reducing the amortized per-iteration cost to $O\left(\frac{mr^2}{V}\right)$. Other operations, such as matrix multiplications involving $P$, are highly optimized on modern GPU architectures and introduce minimal additional overhead.
>
> [1] Chen, Yiming, et al. "Enhancing zeroth-order fine-tuning for language models with low-rank structures."
>
> [2] Yu, Ziming, et al. "Zeroth-order fine-tuning of llms in random subspaces."

---

> > ### Author Rebuttal · Reviewer_ouyd · 2026-04-02
> >
> > I am happy with the author's response, and I will keep my score.

---

> > > ### Author Response · Authors · 2026-04-02
> > >
> > > Dear Reviewer ouyd,
> > >
> > > Thank you very much for acknowledging that our rebuttal has fully addressed your comments. We also greatly appreciate your constructive feedback, and we will surely update the paper accordingly to reflect your constructive comments.
> > >
> > > Thanks,
> > >
> > > Authors

---

### Decision · Program_Chairs · 2026-04-30

**Decision:**

Accept (spotlight)

**Comment:**

All four reviewers recommend acceptance of this work (3x Accept; 1x Strong Accept).
Among their positive aspects of their paper are:
- It is a very well-argued paper for an ambitious and wide position. It is backed with SOTA literature trends, common practices, and examples.
- "With the recent interest in memory-efficient training and gradient-free learning, revisiting ZO optimization in the context of modern deep learning is both timely and relevant."
- "The paper highlights several directions that are underexplored in the literature"

Some of the weaknesses and points where the paper can be improved are:
- Discussion around P5 is a bit unclear
- P4 would benefit from the discussion about the cost of linear algebraic operations.
- Some of the proposed directions are remain speculative.

Overall, all reviewers liked this paper, so I would recommend its acceptance.